# Energy Values of Brewer's Grains and Olive Pomace Waste for Broiler Chickens Determined Using the Regression Method

Carine Beatriz Adams, Otoniel Souza, Jessica Cristina Agilar, Geovana Muller, Beatriz Rodrigues and Catarina Stefanello *





Department of Animal Science, Federal University of Santa Maria, Santa Maria 97105-900, Brazil; carine.adams@acad.ufsm.br (C.B.A.); otoniel.felix@acad.ufsm.br (O.S.); jessica.agilar@acad.ufsm.br (J.C.A.); geovana.muller@acad.ufsm.br (G.M.); beatriz.nascimento@acad.ufsm.br (B.R.)
* Correspondence: catarina.stefanello@ufsm.br; Tel.: +55-55-3226-8083

**Abstract:** Alternative ingredients can be adequately used in poultry feeds as long as the energy values and nutrient digestibility have been previously determined. For example, brewer's grains and olive pomace waste, which are residues of the food industry, are potential ingredients of animal feed. This study was conducted to determine the metabolizable energy (ME), nitrogen-corrected ME ($ME_n$), and ileal digestible energy (IDE) of brewer's grains and olive pomace waste for broiler chickens using the regression method. From day 14 to 21, 280 Cobb 500 male broilers were fed 5 experimental diets with 8 replicates of 7 birds each. The broilers were fed a corn-soy reference diet (RD) and 4 test diets (TD), where TD consisted of brewer's grains or olive pomace waste that partly replaced the energy sources in the RD at 10 or 20% and 7.5 or 15%, respectively. The total tract metabolizability and the apparent ileal digestibility of dry matter, N, and energy as well as ME, $ME_n$, and IDE were determined. The ME, $ME_n$, and IDE values (kcal/kg) were 2935, 2785, and 2524 for brewer's grains, respectively, whereas 1778, 1581, and 1394 (kcal/kg) were obtained for the olive pomace waste, respectively. This knowledge can provide useful information that helps to improve the inclusion of alternative ingredients in broilers diets and to formulate accurate feeds to meet broiler requirements.

**Keywords:** broiler; digestibility; feed formulation; metabolizable energy; nutrition

## 1. Introduction

The poultry industry has grown significantly in recent years and become one of the largest food suppliers for the human population. Therefore, understanding the digestive characteristics of animals as well as the adequate energy supply and nutrient digestibility of ingredients is necessary to improve the efficiency of poultry production. Corn and soybean meal (SBM) are the main ingredients of poultry feeds; however, market instability, exchange rate, high demand for grains, and climate challenges have increased the prices of corn and SBM. Therefore, alternative ingredients that partially replace these main sources of protein and energy are necessary to reduce feed costs, improve environmental sustainability, and develop animal production in some regions [1,2].

Breweries are distributed worldwide and the beer industry is a major contributor to the regional development of many cities. After the processing of germinated and dried cereal grains for beer production, some byproducts are generated as wet or dried brewers, brewer's spent grains, and brewery yeast. To produce 100 L of beer, around 20 kg of waste is generated [3], which must be properly disposed of to avoid environmental pollution. Brewer's grains are characterized as a solid residue that is left after the processing of barley, wheat, corn, rice, or sorghum during beer production. Brewer's grains are collected at the end of the mashing process after fermentation. The remaining product is a concentrate of protein and fiber that has become interesting for animal feed.

Olive oil industries are also found worldwide, and recent increases in olive oil consumption have increased the production of residues. Approximately 20% of the olive fruit

(*Olea europaea* L.) results in extra virgin olive oil, and the remaining 80% becomes olive pomace, which is a waste [4]. Once the waste is disposed of in the environment, severe problems occur because of the high organic content of the poorly degradable phenolic compounds that are toxic for vegetables [5]. To improve the sustainability of the olive oil industry, olive pomace waste has been used for animal nutrition, soil fertilization, and waste composting [6].

Brewer's grains and olive pomace waste are low-cost residues of the processing industry. However, these wastes can be costly to transport due to their perishability and bulkiness when wet. As with many other byproducts, the composition and nutritional values of brewer's grains depend on the industrial process, selected cereals, and methods of preservation, whereas the composition and nutritional values of olive pomace waste can vary according to the residual oil, proportion of stones in the mass, fruit maturation, and plant variety [6].

The published research on nutrients and energy values of brewer's grains and olive pomace waste that are fed to broiler chickens is limited. It has been reported that the crude fiber (CF) content in brewer's grains ranged from 14.7 to 19.6% [7,8], having crude protein (CP) levels of 20.3–29.5% [9,10]. The CP content in olive pomace waste ranged from 5.8 to 10% [2,11], whereas the CF levels were reported to be 23.5–27.6% [8,12]. No above-mentioned studies have determined the energy and nutrient digestibility of these ingredients in animals. Finally, tables of nutrient and ingredient composition [8,13,14] did not include the digestibility of nutrients and energy values of brewer's grains or olive pomace waste for poultry; however, FEEDTABLE [15] reported metabolizable energy at 2270 kcal/kg and 1130 kcal/kg for brewer's grains and olive pomace waste, respectively.

Although the chemical composition of brewer's grains and olive pomace waste has been obtained, no published material has been found demonstrating the digestibility and energy values of these ingredients for broilers, which are needed to improve their utilization and to optimize diet formulation. Therefore, the objective of the present study was to determine the metabolizable energy (ME), nitrogen-corrected ME ($ME_n$), and ileal digestible energy (IDE) of brewer's grains and olive pomace waste for broilers using the regression method. The hypothesis of this study was that brewer's grains have higher ME than olive pomace waste.

## 2. Materials and Methods

The procedures involving the healthcare and management of birds were approved by the Animal Care and Use Committee of the Federal University of Santa Maria (Santa Maria, RS, Brazil).

A total of 320 one-day-old male broiler chicks (Cobb 500), vaccinated for Marek's disease were obtained from a commercial hatchery (Agrogen, Montenegro, RS, Brazil). The birds were weighed before placement (42 g average body weight) and randomly allocated in wire battery cages (0.8 m × 0.4 m). The cages were equipped with one feeder and two nipple drinkers in a climate-controlled room. The birds had ad libitum access to a common starter feed (Table 1) from day 1 to 14. The lighting was continuous until day 21 and the average temperature was controlled to provide comfort throughout the study.

**Table 1.** Ingredient and nutrient composition of the starter diet that was fed from day 1 to 14 and the experimental diets that were fed from day 14 to 21.

| Item | Starter Diet | RD [1] | Brewer's Grains | | Olive Pomace Waste | |
|---|---|---|---|---|---|---|
| | Day 1 to 14 | 0% | 10% | 20% | 7.5% | 15% |
| Ingredients, % | | | | | | |
| Corn | 50.99 | 49.63 | 44.40 | 39.17 | 45.71 | 41.79 |
| Soybean meal | 41.68 | 40.19 | 35.96 | 31.72 | 37.01 | 33.84 |
| Soybean oil | 3.91 | 5.10 | 4.56 | 4.03 | 4.70 | 4.29 |
| Dicalcium phosphate | 0.99 | 1.89 | 1.89 | 1.89 | 1.89 | 1.89 |

Table 1. *Cont.*

| Item | Starter Diet | RD [1] | Brewer's Grains | | Olive Pomace Waste | |
|---|---|---|---|---|---|---|
| | Day 1 to 14 | 0% | 10% | 20% | 7.5% | 15% |
| Limestone | 1.20 | 0.94 | 0.94 | 0.94 | 0.94 | 0.94 |
| Salt | 0.52 | 0.52 | 0.52 | 0.52 | 0.52 | 0.52 |
| DL-Methionine | 0.33 | 0.36 | 0.36 | 0.36 | 0.36 | 0.36 |
| L-Lysine HCl | 0.13 | 0.13 | 0.13 | 0.13 | 0.13 | 0.13 |
| L-Threonine | 0.03 | 0.06 | 0.06 | 0.06 | 0.06 | 0.06 |
| Vit. and min. premix [2] | 0.20 | 0.18 | 0.18 | 0.18 | 0.18 | 0.18 |
| Celite [3] | - | 1.00 | 1.00 | 1.00 | 1.00 | 1.00 |
| Brewer's grains | - | 0.0 | 10.00 | 20.00 | 0.0 | 0.0 |
| Olive pomace waste | - | 0.0 | 0.0 | 0.0 | 7.50 | 15.00 |
| Calculated energy and nutrient composition, % or as shown | | | | | | |
| ME, kcal/kg | 3000 | 3025 | | | | |
| Crude protein (CP) | 23.79 | 22.68 | | | | |
| Calcium | 1.00 | 0.96 | | | | |
| Phosphorus | 0.83 | 0.79 | | | | |
| Total amino acids | | | | | | |
| Arginine | 1.32 | 1.31 | | | | |
| Cysteine | 0.32 | 0.30 | | | | |
| Histidine | 0.67 | 0.59 | | | | |
| Isoleucine | 0.79 | 0.81 | | | | |
| Leucine | 1.89 | 1.62 | | | | |
| Lysine | 1.32 | 1.15 | | | | |
| Methionine | 0.61 | 0.57 | | | | |
| Phenylalanine | 1.20 | 0.98 | | | | |
| Threonine | 0.82 | 0.80 | | | | |
| Valine | 0.88 | 0.87 | | | | |
| Analyzed CP, % | 24.0 | 22.4 | 22.1 | 22.0 | 20.0 | 18.7 |
| Analyzed GE [4], % | 4005 | 4084 | 4163 | 4233 | 4193 | 4313 |

[1] RD = reference diet. [2] Composition per kilogram of feed: copper, 10 mg; iodine, 0.7 mg; iron, 40 mg; manganese, 80 mg; selenium, 0.25 mg; zinc, 80 mg; biotin, 0.1 mg; cyanocobalamin, 0.015 mg; niacin, 35 mg; folic acid, 1.5 mg; pyridoxine, 3.8 mg; pantothenic acid, 12 mg; riboflavin, 6 mg; thiamine, 2 mg; vitamin A, 9000 IU; vitamin $D_3$, 2500 UI; vitamin E, 20 IU; vitamin $K_3$, 2.5 mg. [3] Indigestible marker (Celite, Celite Corp., Lompoc, CA, USA). [4] GE = gross energy.

On day 14, 280 broilers were sorted by weight (557 g $\pm$ 7 g) and distributed into 40 cages. From day 14 to 21, the birds were fed 5 experimental diets, having 8 replicates of 7 birds each. During the experimental period, the broilers also had ad libitum access to water and mash feeds. Celite at 1% was used as an indigestible marker.

In this study, the regression method was used and dietary treatments consisted of a reference diet (RD) and four test diets (TD). A corn-soy feed was prepared as an RD in which soybean oil, soy, and corn were used as the energy-yielding sources (Table 1). A total of four TD were obtained by supplementing the RD with 10 or 20% of brewer's grains and 7.5 or 15% of olive pomace waste. The test ingredients were added to partly replace the energy sources in the TD, then the ratio of soybean oil, SBM, and corn across the experimental diets were maintained the same. These ratios were 7.9:1, 9.7:1, and 1.2 for SBM: soybean oil, corn: soybean oil, and corn: SBM, respectively.

Brewer's grains were obtained from a local beer industry (Santamate Indústria e Comercio, Santa Maria, RS, Brazil), and the product was collected at the end of the mashing process after fermentation in the beer production. The olive pomace waste was obtained from an olive oil agroindustry (Olivas do Sul, Cachoeira do Sul, RS, Brazil). The material was collected right after the extraction of olive oil by biphasic centrifugation and stored at −20 °C. Both the ingredients were wet on receipt, dried in a forced air oven at 55 °C for 3 days (Marconi, model MA 035, Piracicaba, SP, Brazil), and ground in a knife mill with a 2 mm sieve. The brewer's grains and olive pomace waste were analyzed to determine the chemical composition, gross energy (GE), and total amino acids.

### 2.1. Experimental Procedures

On day 14, the chicks were weighed into groups of 7 birds per cage. The bird and feeder weights that were averaged by cage were recorded on days 14 and 21. The excreta was collected twice daily on plastic sheeting from day 19 to 20 being mixed, pooled by cage, and stored at $-20\ ^{\circ}$C until analysis. The ileal digesta was collected from all birds on day 21 after euthanasia by cervical dislocation where the final 2/3 distal ileum were flushed with distilled water into plastic containers and immediately frozen until drying. Excreta and ileal digesta were dried in a forced air oven at $55\ ^{\circ}$C. The feeds and samples of digesta and excreta were ground to pass a 0.5-mm screen in a grinder.

### 2.2. Chemical Analysis

Dry matter (DM) analysis of the feeds, excreta, and ileal digesta was performed after oven drying the samples at $105\ ^{\circ}$C for 16 h (method 934.01; [16]). Nitrogen (N) was determined using the dry combustion method (method 972.43; [16]) in a CN analyzer (Thermo-Finnigan Flash EA 1112, Waltham, MA, USA). The acid insoluble ash concentration in the diet, ileal digesta, and excreta samples were determined according to Choct and Annison [17]. The feeds, ileal digesta, and excreta samples were also analyzed for gross energy (GE) in an adiabatic bomb calorimeter (Parr Instrument Company, 6400, Moline, IL, USA).

### 2.3. Calculations

This methodology has been used to determine the energy values of ingredients for non-ruminant animals. Through the regression method, the ileal digestibility and total tract metabolizability coefficients (C) of energy and nutrients were calculated as previously reported by Bolarinwa and Adeola [18] and Dalmoro et al. [19]. These coefficients were calculated as $C = 1 - [(C_d/C_o) \times (E_o/E_d)]$, where $C_d$ is the concentration of celite in the diet, $C_o$ is the concentration of celite in the excreta or ileal digesta, $E_o$ is the concentration of energy or nutrient in the excreta or ileal digesta, and $E_d$ is the concentration of energy or nutrient in the diet.

The ileal digestible energy (in kcal/kg of DM) of the diet was calculated as the product of C and the GE concentration (in kcal/kg) of the diet. The ME (in kcal/kg) of the diet was calculated as the product of C and the GE concentration (in kcal/kg) of the diet. The metabolizable energy was corrected to zero N retention ($ME_n$) using a factor of 0.03439248 kcal/g [20]. There is further description on the calculation of ME, $ME_n$, or IDE coefficients for the reference diet, test diets, and test ingredient in the publications by Stefanello et al. [21], Zhang and Adeola [22], and Adeola and Kong [23]. The product of the coefficients of IDE, ME, or $ME_n$ at each level of test ingredient substitution rate (brewer's grains, 10 or 20% and olive pomace waste, 7.5 or 15%), kg of test ingredients intake (product of 0.1, 0.2 or 0.075, 0.15 dry feed intake), and the GE of the test ingredients are the test ingredient-associated ME, $ME_n$, or IDE in kcal [18,22,23].

### 2.4. Statistical Analysis

The data were analyzed using the GLM procedure of SAS [24]. Regressions of the test ingredient-associated ME, $ME_n$, or IDE intake in kcal against kg of test ingredient intake for cage of broilers were conducted using linear regression following SAS statements according to Bolarinwa and Adeola [18]. The effects of increasing levels of brewer's grains and olive pomace waste in the assay diets were compared using linear and quadratic contrasts. Statistical significance was determined at $p < 0.05$.

## 3. Results

The analyzed nutrient composition of the brewer's grains and olive pomace waste is presented in Table 2. Both the ingredients were selected because they are food industry residues that potentially pollute the environment. Brewer's grains and olive pomace are wastes that are spread around the world but they also have a local representative

importance, which contributes to the regional development when used as ingredients for animal nutrition. The crude protein, ash, calcium, and phosphorus contents were higher in the brewer's grains compared to the olive pomace waste. On the other hand, the olive pomace waste presented higher gross energy and crude fiber than the brewer's grains. The high gross energy is due to the high ether extract content, whereas the olive stones result in high crude fiber concentration.

**Table 2.** Analyzed chemical composition and the total amino acids of the brewer's grains and olive pomace waste.

| Item, % | Brewer's Grains | Olive Pomace Waste |
|---|---|---|
| Dry matter | 92.71 | 95.57 |
| Gross energy, kcal/kg | 4843 | 5853 |
| Crude protein (N $\times$ 6.25) | 24.07 | 6.20 |
| Ether extract | 6.51 | 21.93 |
| Crude fiber | 17.11 | 37.68 |
| Acid detergent fiber | 20.59 | 45.50 |
| Neutral detergent fiber | 60.90 | 54.81 |
| Ash | 6.34 | 2.51 |
| Calcium | 0.20 | 0.16 |
| Phosphorus | 0.49 | 0.14 |
| Indispensable amino acids | | |
| Arginine | 1.07 | 0.30 |
| Histidine | 0.48 | 0.12 |
| Isoleucine | 0.71 | 0.22 |
| Leucine | 1.50 | 0.41 |
| Lysine | 0.86 | 0.03 |
| Methionine | 0.32 | 0.05 |
| Phenylalanine | 1.18 | 0.28 |
| Threonine | 0.72 | 0.21 |
| Valine | 0.96 | 0.27 |
| Dispensable amino acids | | |
| Alanine | 1.01 | 0.26 |
| Aspartic acid | 1.46 | 0.63 |
| Cysteine | 0.32 | 0.10 |
| Glutamic acid | 4.37 | 0.63 |
| Glycine | 0.91 | 0.29 |
| Proline | 2.19 | 0.26 |
| Serine | 1.02 | 0.27 |
| Tyrosine | 0.69 | 0.12 |

The inclusion of brewer's grains or olive pomace waste in the reference diet did not result in quadratic effects for all the evaluated responses. Only linear decreases ($p < 0.05$) were observed in the performance variables (Table 3). There was a linear decrease ($p < 0.05$) on the final body weight, weight:gain, feed intake, and gain:feed when the brewer's grains or olive pomace waste were included in the reference diet.

The total tract metabolizability coefficients, ME, and $ME_n$ as well as the ileal digestibility coefficients and IDE of the experimental diets are presented in Table 4. Linear decreases ($p < 0.05$) were observed on ME and IDE of the diets with increasing brewer's grains. There were no effects ($p > 0.05$) on the metabolizability or ileal digestibility coefficients of DM and N when brewer's grains were included. The addition of olive pomace waste linearly decreased ($p < 0.05$) the ileal digestibility and metabolizability coefficients of DM, N, and energy as well as linearly decreased ($p < 0.05$) ME, $ME_n$, and IDE.

**Table 3.** Growth performance of broilers that were fed experimental diets containing brewer's grains or olive pomace waste from day 14 to 21.

| Item | RD [1] | Brewer's Grains | | Olive Pomace Waste | | SEM | p-Value | | | |
| | | | | | | | Brewer's Grains | | Olive Pomace Waste | |
| | | 10% | 20% | 7.5% | 15% | | L [2] | Q [2] | L | Q |
|---|---|---|---|---|---|---|---|---|---|---|
| Initial BW, g | 562 | 558 | 550 | 561 | 557 | 2.0 | 0.416 | 0.837 | 0.105 | 0.762 |
| Final BW, g | 1120 | 1050 | 864 | 1078 | 1034 | 16.2 | 0.018 | 0.777 | 0.001 | 0.092 |
| Weight gain, g | 558 | 492 | 314 | 517 | 477 | 15.4 | 0.022 | 0.748 | 0.001 | 0.097 |
| Feed intake, g | 742 | 687 | 582 | 740 | 709 | 11.1 | 0.009 | 0.151 | 0.001 | 0.200 |
| Gain:Feed, g:kg | 753 | 716 | 532 | 700 | 673 | 15.3 | 0.007 | 0.277 | 0.001 | 0.289 |

[1] RD = Reference diet. The RD was isonitrogenous and isocaloric, however test diets used brewer's grains or olive pomace waste to replace soybean oil, soybean meal, and corn at the same ratio. [2] Linear and quadratic contrasts for brewer's grains or olive pomace waste ($n = 40$).

**Table 4.** Apparent ileal digestibility and total tract metabolizability of DM, nitrogen, and energy of broilers that were fed experimental diets containing brewer's grains or olive pomace waste from day 14 to 21.

| Item | RD [1] | Brewer's Grains | | Olive Pomace Waste | | SEM | p-Value | | | |
| | | | | | | | Brewer's Grains | | Olive Pomace Waste | |
| | | 10% | 20% | 7.5% | 15% | | L [2] | Q [2] | L | Q |
|---|---|---|---|---|---|---|---|---|---|---|
| *Ileal digestibility* | | | | | | | | | | |
| Dry matter coefficient | 0.70 | 0.69 | 0.67 | 0.67 | 0.62 | 0.014 | 0.235 | 0.245 | 0.001 | 0.149 |
| Nitrogen coefficient | 0.86 | 0.85 | 0.85 | 0.81 | 0.77 | 0.017 | 0.889 | 0.898 | 0.010 | 0.746 |
| Energy coefficient | 0.73 | 0.71 | 0.68 | 0.70 | 0.64 | 0.014 | 0.036 | 0.390 | 0.001 | 0.190 |
| IDE [3], kcal/kg | 3345 | 3312 | 3196 | 3319 | 3070 | 0.07 | 0.005 | 0.688 | 0.001 | 0.726 |
| *Total tract metabolizability* | | | | | | | | | | |
| Dry matter coefficient | 0.69 | 0.68 | 0.67 | 0.66 | 0.61 | 0.009 | 0.526 | 0.210 | 0.001 | 0.174 |
| Nitrogen coefficient | 0.72 | 0.71 | 0.65 | 0.64 | 0.60 | 0.029 | 0.056 | 0.110 | 0.001 | 0.104 |
| Energy coefficient | 0.75 | 0.73 | 0.72 | 0.71 | 0.67 | 0.009 | 0.092 | 0.101 | 0.002 | 0.102 |
| Nitrogen–correct energy coefficient | 0.70 | 0.68 | 0.67 | 0.67 | 0.63 | 0.008 | 0.154 | 0.249 | 0.001 | 0.118 |
| ME [4], kcal/kg DM [5] | 3429 | 3404 | 3372 | 3368 | 3185 | 0.04 | 0.040 | 0.233 | 0.009 | 0.194 |
| $ME_n$, kcal/kg DM | 3191 | 3175 | 3166 | 3184 | 3021 | 0.04 | 0.396 | 0.971 | 0.045 | 0.195 |

[1] RD = Reference diet. The RD was isonitrogenous and isocaloric, however test diets used brewer's grains or olive pomace waste to replace soybean oil, soybean meal, and corn at the same ratio. [2] Linear and quadratic contrasts for brewer's grains or olive pomace waste ($n = 40$). [3] IDE = ileal digestible energy. [4] ME = apparent metabolizable energy. [5] DM = dry matter.

The regression of apparent ME, apparent $ME_n$, and IDE intake that is associated with brewer's grains or olive pomace waste for broilers is presented in Table 5. The obtained regressions were: $ME = 2935x + 7739$, $ME_n = 2785x + 11,302$, and $IDE = 2524x + 16,366$ for the diet with brewer's grains; whereas $ME = 1778x + 7739$, $ME_n = 1581x + 11,302$, and $IDE = 1394x + 16,366$ were obtained for the diet with olive pomace waste substitution ($p < 0.05$). Thus, the ME and $ME_n$ values for brewer's grains were 2935 and 2785 kcal/kg DM, respectively, while the ME and $ME_n$ for olive pomace waste were 1778 and 1581 kcal/kg DM, respectively. The IDE for the brewer's grains was 2524 kcal/kg DM and for the olive pomace it was 1394 kcal/kg DM (Table 5).

**Table 5.** Regressions analysis relating the test ingredient-associated energy intake to the intake of brewer's grains or olive pomace waste.

| Item | Regression Equation | SE of Intercept [1] | SE of Slope | *p*-Value | $r^2$ |
|---|---|---|---|---|---|
| | | Brewer's grains | | | |
| ME [1] | Y = 2935x + 7.739 | 7.97 | 112.98 | 0.001 | 0.948 |
| ME$_n$ | Y = 2785x + 11.302 | 8.33 | 118.12 | 0.001 | 0.940 |
| IDE [2] | Y = 2524x + 16.366 | 12.08 | 171.28 | 0.001 | 0.855 |
| | | Olive pomace waste | | | |
| ME | Y = 1778x + 7.739 | 7.97 | 99.57 | 0.001 | 0.948 |
| ME$_n$ | Y = 1581x + 11.302 | 8.33 | 95.24 | 0.001 | 0.940 |
| IDE | Y = 1394x + 16.366 | 12.08 | 144.37 | 0.001 | 0.855 |

[1] ME = apparent metabolizable energy. [2] IDE = Ileal digestible energy.

## 4. Discussion

Alternative ingredients have been used in feed formulations for broilers due to the high prices of corn and soybean meal, which are the main energy and protein sources for poultry. Furthermore, the use of byproducts with potential interest in animal nutrition is considered an opportunity to increase the economic and environmental sustainability of livestock farms and agribusinesses [2]. Brewer's grains and olive pomace waste are two residues that have been produced at a large scale, besides the potential of using these ingredients for animal nutrition. Thus, determining the energy values of alternative ingredients for broilers becomes necessary to formulate diets with adequate levels, which meet the nutritional requirements for their maximum performance. In the present study, the energy values (ME, ME$_n$, and IDE) of brewer's grains and olive pomace waste were determined for broilers.

The regression of energy or the nutrient digestibility against the proportions of energy or nutrients that were replaced and extrapolation to 100% replacement was used to determine the digestibility of the components in the test ingredients [23], and then the energy values can be determined. In the regression analysis, a RD is fed to one group of broilers, and at least two TD are fed to other broilers, with energy components in the RD being partially replaced by two levels of the test ingredient. The coefficient of energy digestibility of the test ingredient in each TD can be calculated using the equations; then the test ingredient-associated energy intake in kcal can be calculated and regressed against kg of the test ingredient intake for broilers to generate intercepts and slopes, where the slope is the energy in kcal/kg of DM of the test ingredients [18].

The chemical composition of the brewer's grains can be variable due to several factors such as cereal types, soil, harvest period, brewing processes, and added adjuvants during brewing [25]. The brewer's grains had 24.0% CP, 17.1% CF, 6.51% ether extract (EE), and 4843 kcal/kg GE in the present study. The analyzed CP content is in agreement with findings by Senthilkumar et al. [7] and Carvalho et al. [26], with brewer's grains having 24.3% and 24.4% CP, respectively, whereas Silva et al. [9] and NRC [27] reported 20.3% and 26.5% CP, respectively. The CF content of the brewer's grains was reported as 14.7% [8] and 19.6% [7]. Additionally, Senthilkumar et al. [7], NRC [27] and Tesser [28] reported 3543, 4805, and 4527 kcal/kg of GE, respectively. These studies did not determine the metabolizable energy or digestibility of brewer's grains for animals.

Babarinde et al. [29] evaluated dried brewer's grains in broiler diets at 0 and 15% and did not observe differences on weight gain, carcass, and organ weights of broilers. Similarly, Denstadli et al. [3] fed broilers with 0, 10, 20, 30, and 40% dry brewer's grains from day 12 to 33 and found no difference in weight gain with up to 20% of brewer's grains. From 30 to 40% inclusion, these authors observed a linear decrease in body weight. Parpinelli et al. [30] also did not observe effects of dry brewer's grains on broiler performance using inclusions at 0, 2, 4, 6, 8, and 10% for broilers until 21 days of age.

In the case of olive pomace waste, the composition of residual oil and water contents, proportion of stone parts in the mass, fruit maturation, and plant variety have been

indicated as causes of variability in its composition [6]. In the current study, the olive pomace waste had 6.2% CP, 37.8% CF, 21.93% EE, and 5853 kcal/kg GE. Guido et al. [11] found similar values at 5.8% CP and 5828 kcal/kg GE. Guerreiro et al. [2] reported 10% CP and 5144 kcal/kg GE, whereas Pappas et al. [12] obtained 8.6% CP and 27.6% CF of olive pomace waste.

The olive pomace waste presented lower CP content and higher CF and GE compared to the brewer's grains. Although the tested ingredients presented higher GE, Ca, and fat contents compared to corn, their high fiber content probably can affect nutrient digestibility and energy utilization. Crude fiber can negatively affect the feed intake and digestibility of ingredients for broilers when the levels are higher than 3.5% [31]. According to Mateos et al. [32], the inclusion of up to 3% of insoluble fiber in diets of young chicks based on corn and SBM might improve the development of the digestive tract and growth performance. It has been demonstrated that the inclusion of moderate amounts of fiber sources increased HCl, bile acids, and enzyme secretion [33,34] as well as improved the functionality of the gastrointestinal tract. Then, fiber sources have become more studied for broilers; however, the digestibility is very limited if the diets are formulated with high CF levels for broilers.

Papadomichelakis et al. [35] suggested that levels up to 5% of olive pomace waste can be used for broilers in the finisher phase, because after 28 days of age, the digestive tract is developed and also adapted to digest fibers. El-Hackemi et al. [36] did not find differences in the weight gain and carcass weight of broilers that were fed a control diet or diets with 5, 10, or 15% olive pomace waste, demonstrating the possibility of using up to 15% olive pomace in broilers feeds. Similarly, Sayehban et al. [37] evaluated the performance of broilers that were fed 0, 5, and 10% olive pomace, and did not find differences on growth performance and indicated that its inclusion can be up to 10% of feed.

Although some previous research reported that brewer's grains or olive pomace waste could be included in the diets for broilers reaching levels at 10 or 15% without decreasing the performance, it is necessary firstly obtain ME values and nutrient digestibility to improve the usage of both ingredients and to formulate more accurate diets. Considering the ingredient composition and the obtained ME values in the current study, these ingredients seem to be promising sources of fiber and protein for broilers, but especially for pullets, laying, or breeder hens. Further research is needed to evaluate the effects of increasing levels of brewer's grains and olive pomace waste for hens.

Investigating the nutritional composition, sensory characteristics, color, and physical properties of meat and eggs are essential to guarantee the benefit of using residues in the nutrition of the birds. Authors have reported that phenolic compounds that are extracted from the olive industry might be beneficial to birds also through their antimicrobial activity, antioxidant and/or immunomodulatory effects [38], and anti-inflammatory functions [39].

As the test diets were not isocaloric and isonitrogenous because of the regression method, it was expected linear or quadratic effects when increasing levels of the tested ingredients replaced the energy-yielding sources (corn, soy, and soy oil). Thus, regression equations could be estimated considering the ingredient-associated energy intake, since the main objective of this work was not to evaluate the broilers' performance. According to Kong and Adeola [23], in the energy utilization studies, the classical ME assay and the precision-fed method are applicable for poultry. Digestibility trials with poultry have used a single level of substitution of a test ingredient such that the reference diet has been used to estimate the energy content [40,41] or at least two levels of substitution of a reference diet component in the regression method [42].

In the literature, data on ME, $ME_n$, and IDE of brewer's grains and olive pomace waste are still scarce and determining the energy values of these ingredients is necessary for greater precision in feed formulations. The published $ME_n$ of brewer's grains and olive pomace waste for poultry were reported by FEEDTABLE [15] at 2270 kcal/kg and 1130 kcal/kg, respectively. The ME, $ME_n$, and IDE values of brewer's grains that were found in the current study were 2935, 2785, and 2524 kcal/kg, respectively. The val-

ues of ME, MEn, and IDE of olive pomace found in the present study were 1778, 1581, and 1394 kcal/kg, respectively. The tables of nutrient and ingredient composition [8,13,14] did not present digestibility of nutrients and energy values of brewer's grains or olive pomace waste for poultry.

## 5. Conclusions

The energy values of brewer's grains and olive pomace waste were determined using the regression method. The brewer's grains presented higher energy values if compared to olive pomace waste and both ingredients provided a considerable amount of fibers. The ME, MEn, and IDE values were 2935, 2785, and 2524 kcal/kg for the brewer's grains, respectively, whereas 1778, 1581, and 1394 kcal/kg were obtained for the olive pomace waste, respectively. Besides these ingredients have variable nutrient composition as many plant byproducts, the obtained energy values are valuable to formulate accurate diets for broilers. Further studies are needed to evaluate the inclusion of increasing levels of brewer's grains or olive pomace waste in the diets for pullet, laying, and breeder hens.

**Author Contributions:** Conceptualization, C.S. and C.B.A.; methodology, C.B.A.; validation, O.S., C.B.A. and B.R.; formal analysis, C.B.A.; investigation, C.B.A., J.C.A., G.M. and B.R.; resources, C.S.; data curation, C.B.A.; writing—original draft preparation, C.S. and C.B.A.; writing—review and editing, C.S; supervision, C.S.; project administration, C.S. All authors have read and agreed to the published version of the manuscript.

**Funding:** This research received no external funding.

**Institutional Review Board Statement:** This study was conducted according to the guidelines of the Declaration of Helsinki, and was approved by the Ethics and Research Committee of the Federal University of Santa Maria, Santa Maria, RS, Brazil (protocol code: 5404280717, 15 August 2017).

**Informed Consent Statement:** Not applicable.

**Data Availability Statement:** Data that were presented in this study are available on request from the corresponding author.

**Acknowledgments:** The authors wish to thank Coordenação de Aperfeiçoamento de Pessoal de Nível Superior (CAPES–Brasilia, DF, Brazil) for the partial scholarship paid to the first author.

**Conflicts of Interest:** The authors declare no conflict of interest.

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
