# Peer review of "Energy Values of Brewer’s Grains and Olive Pomace Waste for Broiler Chickens Determined Using the Regression Method"

_agriculture, doi:10.3390/agriculture12040444_

Round 1
Reviewer 1 Report
The study appears to have been conducted with due care and the paper is well written. Some moderate amendments are suggested to further improve the clarity.
L109: Both test products are known to be highly variable in composition and therefore the ME. More details should be added of their sources, production etc.
L134: the statement is incorrect. Regression is the least used of the three methods of ME determination (direct, difference & regression) – rephrase
DISCUSSION SECTION
Comment: any limitations of the regression method – only 3 levels were used
Other quoted ME values: what methods were used in the determination. Certainly not by regression. Add these details comparison may be instructive
What was the significance of IDE measurement ?. comment
Add a comment – these products require drying, with added cost. That will be key deterrent to their practical use
Author Response
Manuscript Number: agriculture-1622749
Dear Reviewer 1
This is a response for the reviewer 1 comments on the above cited manuscript.
I have attached a new version of the manuscript which has addressed comments and suggestions from the three reviewers. We used track changes to show the changes we have made in the revised manuscript.
Thank you! We have been putting our best efforts to correct this manuscript for publication. We understand that your comments are important, so we followed your thoughts and deleted/included information in this manuscript as recommended.
Reviewer #1: The study appears to have been conducted with due care and the paper is well written. Some moderate amendments are suggested to further improve the clarity.
L109: Both test products are known to be highly variable in composition and therefore the ME. More details should be added of their sources, production etc.
Response: Completed. It was described in the introduction and discussion section; however, we included more information in the material and methods as requested.
L134: the statement is incorrect. Regression is the least used of the three methods of ME determination (direct, difference & regression) – rephrase
Response: Done as requested. This sentence was modified.
DISCUSSION SECTION
Comment: any limitations of the regression method – only 3 levels were used
Response: Done as requested. More information was included.
Other quoted ME values: what methods were used in the determination. Certainly not by regression. Add these details comparison may be instructive
What was the significance of IDE measurement ?. comment
Response: Done as requested. More information was included in the discussion.
The IDE is not usual for diet formulation i.e.; however, as ileal digesta was collected in the present study, IDE was also determined because it represents another method of energy measurement.
Add a comment – these products require drying, with added cost. That will be key deterrent to their practical use
Response: This was described in the introduction. As small quantities of the test ingredients were used in the present study, the cost with drying was not considered.
Thank you!
Reviewer 2 Report
I suggest to improve the following formulations in the text:
- Line 163-165: Paragraph Results. “It was previously known …. “ This sentence should be in the section Discussion or Introduction.
- Section Results: Why didn’t you provide “n=…” in tables 3 and 4? Please describe all the tables correctly.
- Section Results – Tables: Each time, all abbreviations from the table should be explained below the table. Please revise this.
- Section Discussion: Rather than simply quoting statements from the work of others authors, you may offer more in depth analyses of the novel knowledge, and discuss this in the context of your research. Please try to do this.
Author Response
Manuscript Number: agriculture-1622749
Dear Reviewer 2,
This is a response for the reviewer 2 comments on the above cited manuscript.
I have attached a new version of the manuscript which has addressed comments and suggestions from the three reviewers. We used track changes to show the changes we have made in the revised manuscript.
Thank you! We understand that your comments are important, so we followed your thoughts and deleted/included information in this manuscript as recommended. We also improved the results/discussion and tables to provide appropriate emphasis and inspiring statements.
Reviewer #2: I suggest to improve the following formulations in the text:
1. Line 163-165: Paragraph Results. “It was previously known …. “ This sentence should be in the section Discussion or Introduction.
Response: Done as requested. This sentence was modified.
2. Section Results: Why didn’t you provide “n=…” in tables 3 and 4? Please describe all the tables correctly.
Response: Done as requested. “n” was included.
3. Section Results – Tables: Each time, all abbreviations from the table should be explained below the table. Please revise this.
Response: Done as requested.
4. Section Discussion: Rather than simply quoting statements from the work of others authors, you may offer more in depth analyses of the novel knowledge, and discuss this in the context of your research. Please try to do this.
Response: Completed. As the objective of this study was to determine energy values of ingredients, there is a limitation on discussion. However, we included statements to offer more in depth analyses.
Thank you!
Reviewer 3 Report
To Authors
The article I reviewed concerns an interesting and needed topic, which is the replacement of corn meal and soybean meal (SBM), i.e. the main ingredients of poultry feed, with alternative ingredients that replace the previously mentioned sources of protein and energy. Such measures are important from the point of view of the costs of feed, the improvement of the quality of the natural environment and the sustainable development of animal production in some regions. Brewed grains as high in protein and fiber, as well as olive pomace waste, can be an interesting ingredient in animal nutrition. Importantly, neutralizing olive pomace waste in the environment favors the accumulation of a large amount of poorly decomposing phenolic compounds and other potentially toxic organic compounds acidifying the soil, which disrupts the production of other plants, especially vegetables. I found the idea of ​​the research interesting and, it's main practical. However, the article requires some elaboration. I please create a research hypothesis and draw conclusions based on it. It is very important. This type of research also gives the opportunity to propose conclusions of an application nature, please do it as well. I have included the remaining comments in the review.
Line 9-22: Why were potentially toxic substances not analysed? Fungi, yeast and bacteria are common microorganisms in the malting and brewing cereals. Unfortunately, some microorganisms and their harmful metabolites - toxins, which due to their thermostable properties can be easily transferred to malting and brewing by-products, can be very dangerous to the health of animals. Literature reports that during the storage of such raw materials, aflatoxin, fumonisins, zearalenone and other mycotoxins appear in almost 100% of samples. The level of mycotoxins increases during the storage of the raw material. In addition, some of these toxins are hormonally active. Please clarify this in the review.
Line 69-74: Why wasn't a research hypothesis created? Please complete this.
Line 80: Why were only male birds used? Was it dictated by something?
Line 89: Why was experimental feed used only during this period?
Line 132: What is this analytical technique? Please specify in the manuscript.
Line 289-297: These are not the correct conclusions of this research. They are a duplication of the results. Please create a research hypothesis and then draw conclusions. Practical conclusions are also worth giving.
Author Response
Manuscript Number: agriculture-1622749
Dear Reviewer 3,
This is a response for the reviewer 3 comments on the above cited manuscript.
I have attached a new version of the manuscript which has addressed comments and suggestions from the three reviewers. We used track changes to show the changes we have made in the revised manuscript.
Reviewer #3: The article I reviewed concerns an interesting and needed topic, which is the replacement of corn meal and soybean meal (SBM), i.e. the main ingredients of poultry feed, with alternative ingredients that replace the previously mentioned sources of protein and energy. Such measures are important from the point of view of the costs of feed, the improvement of the quality of the natural environment and the sustainable development of animal production in some regions. Brewed grains as high in protein and fiber, as well as olive pomace waste, can be an interesting ingredient in animal nutrition. Importantly, neutralizing olive pomace waste in the environment favors the accumulation of a large amount of poorly decomposing phenolic compounds and other potentially toxic organic compounds acidifying the soil, which disrupts the production of other plants, especially vegetables. I found the idea of ​​the research interesting and, it's main practical. However, the article requires some elaboration. I please create a research hypothesis and draw conclusions based on it. It is very important. This type of research also gives the opportunity to propose conclusions of an application nature, please do it as well. I have included the remaining comments in the review.
Thank you! We followed your thoughts and deleted/included information in this manuscript as recommended. We also improved the text to provide appropriate emphasis and inspiring statements.
Line 9-22: Why were potentially toxic substances not analysed? Fungi, yeast and bacteria are common microorganisms in the malting and brewing cereals. Unfortunately, some microorganisms and their harmful metabolites - toxins, which due to their thermostable properties can be easily transferred to malting and brewing by-products, can be very dangerous to the health of animals. Literature reports that during the storage of such raw materials, aflatoxin, fumonisins, zearalenone and other mycotoxins appear in almost 100% of samples. The level of mycotoxins increases during the storage of the raw material. In addition, some of these toxins are hormonally active. Please clarify this in the review.
Response: There are no standards of toxic substances for the ingredients and previous publications did not recommend these analyses. Considering the objective of this study and the cost of these analyses, it was not analysed in the current study; however, we included total amino acids composition of the ingredients (table 2). Therefore, as olive pomance waste and brewers grains would be valuable ingredients for pullets and laying hens, all mycotoxins will be analysed in the performance study.
Line 69-74: Why wasn't a research hypothesis created? Please complete this.
Response: Done as requested. Hypothesis was included.
Line 80: Why were only male birds used? Was it dictated by something?
Response: That was a decision. We choose males.
Line 89: Why was experimental feed used only during this period?
Response: The feeding period is determined by the method. A regression method can be used to determine the digestibility of components by having serial proportions of the test ingredient replacing the basal diet. The regression of the digestibility of the component against proportions of the component replaced and extrapolation to 100% replacement is used to determine digestibility of components in test ingredient. Additionally, this method has been used to determine the basal endogenous AA losses including the feeding a highly-digestible and feeding an N-free diet.
Line 132: What is this analytical technique? Please specify in the manuscript.
Response: Response: Done as requested. This information was included, as reviewer 1 also suggested, and in the discussion.
“Direct or difference procedures can be applied in in vivo digestibility studies. In the direct procedure, the test feed ingredient is formulated as the sole source of the component in the test diet. In the substitution procedure, a basal diet is fed to a group of broilers to determine the digestibility of the components. Another group of broilers is fed a test diet with a known proportion of the component from basal diet replacing the test ingredient. The test diet can also be formulated as a basal diet plus a certain quantity of the test ingredient.”
“The regression procedure has also been proven to be a reliable procedure to determine energy and nutrients digestibility. In the regression procedure, a basal diet is fed to one group of broilers, and at least 2 test diets are fed to other broilers, with energy component in the basal diet being partially replaced by 2 levels of the test ingredient feed. The coefficient of energy digestibility of the test ingredient in each test diet can be calculated using the equations; then test ingredient associated ME intake in kilocalories can be calculated and regressed against kilograms of test ingredient intake for broilers to generate intercepts and slopes, where the slope is the ME in kcal/kg of DM of test ingredients.”
Line 289-297: These are not the correct conclusions of this research. They are a duplication of the results. Please create a research hypothesis and then draw conclusions. Practical conclusions are also worth giving.
Response: The objective of the present study was to determine energy values of both ingredients. Then, the main conclusion is the obtained ME, MEn, IDE. Additionally, we indicated in our conclusion that besides these ingredients have variable nutrient composition as many plant byproducts, the obtained energy values are valuable to formulate accurate diets for broilers, which is necessary. Further studies are needed to evaluate inclusion of increasing levels of brewer’s grains or olive pomace waste in diets for pullet, laying, and breeder hens. We think these ingredients would be appropriate for pullets and laying hens.
Thank you!